# Yttrium’s Effect on the Hot Cracking and Creep Properties of a Ni-Based Superalloy Built Up by Additive Manufacturing

**DOI:** 10.3390/ma14051143

**Published:** 2021-02-28

**Authors:** Santhosh Banoth, Thaviti Naidu Palleda, Sota Shimazu, Koji Kakehi

**Affiliations:** Department of Mechanical Systems Engineering, Tokyo Metropolitan University, 1-1, Minami-Osawa, Hachioji-City, Tokyo 192-0397, Japan; shiva.rgukt546@gmail.com (T.N.P.); u15sshimazu@gmail.com (S.S.)

**Keywords:** rare earth element yttrium, hot cracking, selective laser melting, Hastelloy-X, creep

## Abstract

We studied the effects of the rare earth element yttrium (Y) on the hot cracking and creep properties of Hastelloy-X processed by selective laser melting. We used two different alloys to study hot cracking in Hastelloy-X: one with 0.12 mass% yttrium added and one with no yttrium. Y-free Hastelloy-X exhibited less cracks, mainly due to the segregation of Si, W, and C resulting in SiC- and W_6_C-type carbides at the grain boundary and interdendritic regions. On the other hand, more cracks formed in the Y-added Hastelloy-X specimen because of segregation of Y, resulting in the formation of yttrium-rich carbide (YC). Post-heat treatment was conducted at 1177 °C for 2 h, followed by air cooling, to obtain good creep properties. We carried out a creep test along the vertical and horizontal directions. Despite having more cracks, the Y-added as-built Hastelloy-X specimen showed longer creep life and ductility than the Hastelloy-X specimen. This was mainly because of the formation of Y_2_O_3_ and SiO_2_ inside the grains. After solution treatment, the Y-added specimen’s creep life was eight times longer than that of the Y-free solution-treated specimen. This was mainly because of the maintenance of the columnar grain morphology even after solution treatment. In addition, the formation of M_6_C carbides, Y_2_O_3_, and SiO_2_ improved creep life. To summarize the effect of Y, Y addition promoted the formation of cracks, which brought about creep anisotropy; however, it improved creep properties through the stabilization of oxygen and the promotion of discrete carbide precipitation, which prohibited the migration and sliding of grain boundary.

## 1. Introduction

Selective laser melting (SLM) is an advanced technology in additive manufacturing (AM) for the fabrication of metallic components with complex shapes by using layer-by-layer deposition via a high-power laser [1,2,3]. Hastelloy-X is a solid-solution-strengthened Ni-based superalloy with excellent high-temperature oxidation and corrosion resistance, formability, and mechanical properties in the temperature range of 1000–1200 °C. Because of these attributes, it can be applied in aerospace engineering, such as in combustion chambers, cabin heaters, spray bars, and gas turbine engine components [4,5,6]. In 2013, the gas turbine manufacturer Siemens successfully used this material in additive manufacturing to rapidly construct and repair components with Electro-Optical System (EOS) SLM technology. Nevertheless, owing to the extreme temperature gradient and the fast heating and cooling (≈10^6^ K/s) of the SLM process [7], nickel-based superalloys such as Hastelloy-X [8,9], IN718 [10], and CM247LC [11], to name a few, are prone to hot cracking, which degrades their mechanical and physical properties [12].

The main purpose of alloying elements is to improve the mechanical and thermal properties of nickel-based superalloys in order to minimize their susceptibility to hot cracking. Quanquan et al. reported that high-melting-point elements, such as Mo and Cr, result in the formation of high-angle grain boundaries due to the formation of Mo- and Cr-rich carbides at the grain boundaries, eventually resulting in the formation of cracks [13]. The appearance of carbides at the grain boundary has an additional effect of increasing resistance to grain boundary sliding at higher temperatures [14].

Dacian et al. reported the effects of alloying elements such as Mn, Si, and C on hot cracking in Hastelloy-X [15,16]. Lower Mn, Si, and C concentrations can result in fewer cracks from minor microsegregation formation along grain boundaries and interdendritic regions. Dacian et al. also studied the effects of these alloying elements on hot cracking by using the computational thermodynamic approach. Those authors proposed that the quantities of Si and C are the main influences on the cracking mechanism. In contrast, Mn has a negligible effect [17].

The addition of rare earth elements such as Y and Ce is usually the primary consideration in alloy design because of their effects on the strengthening of both grain boundaries and solid solution [18]. Yttrium, a famous rare earth (RE) element, has been applied successfully in many fields such as metallurgy, chemical, and surface engineering. In recent years, yttrium has been added to many alloys, including nickel-based superalloys, to improve their physical and mechanical properties. Zhou et al.’s work showed that the optimal level of yttrium in nickel-based alloys improves the stress-rupture property [19] and the oxidation resistance of nickel-based superalloy [20,21]. Yttrium addition in cast stainless steels improves creep property [22], alumina [23], and Fe–Ni–Cr [24]. However, there has been little study of yttrium’s effects on the microstructure and strength performance of nickel-based superalloys.

In the present study, we fabricated Hastelloy-X using the SLM process in an Ar atmosphere. We added two different yttrium levels (0 and 0.12 mass%) to Hastelloy-X to study yttrium’s effects on hot cracking and creep properties in Hastelloy-X. We also analyzed yttrium’s effects on the microstructure, creep properties, and hot cracking in Hastelloy-X processed by the SLM process.

## 2. Materials and Methods

The exact chemical compositions (in mass%) of Hastelloy-X alloys with 0 and 0.12 yttrium are called HX and HX-a, respectively, and are shown in Table 1. These two specimens were built in the shape of a 45 × 45 × 45 mm cube by using the EOS M290 SLM machine (E.O.S., Robert-Stirling-Ring 1, 82152, Krailling, Bavaria, Germany) in a protective Ar atmosphere using pre-alloyed powders and the same processing parameters. We conducted standard heat treatment for both the HX and HX-a specimens. Solution heat treatment (ST) was performed at 1177 °C for 2 h, followed by air cooling to room temperature.

For the creep test, we sliced the cube into a number of slabs with a thickness of 3.1 mm; from these slabs, the creep test specimens were cut out using an electro-discharge wire cutting machine. The gauge dimensions of each specimen were 19.6 × 2.8 × 3.0 mm. We conducted the creep test under 900 °C/80 MPa conditions. The specimens were polished using SiC emery paper up to grade 1200# followed by diamond paste up to colloidal silica (0.5 µm) using a Struers (Ballerup, Denmark) automatic polishing machine. All the specimens were then washed with ethanol in an ultrasonic bath for 10 min. We etched the specimens with 20% phosphoric acid + 80% water solution to observe the molten pool boundaries. The microstructural observation was conducted using an optical microscope (OM; Olympus Corp. Tokyo, Japan), a scanning electron microscope (SEM; Hitachi, Ltd., Tokyo, Japan), energy-dispersive spectroscopy (EDS) (S-3700N type EDS equipment manufactured by Horiba Seisakusho Co., Ltd., Kyoto, Japan), and field emission electron microscopy (FE-SEM) (JSM-7100, JEOL, Tokyo, Japan) attached to an EDS (EDAX AMETEX 9424). Image J software (64-bit Java 1.8.0_172) was used to analyze crack fraction and porosity measurements.

## 3. Results

### 3.1. Microstructure Observation

#### 3.1.1. As-Built Specimens

Figure 1 shows the HX and HX-a specimens’ optical microstructures in the as-built condition. All were manufactured using the same processing parameters. The HX-a specimen exhibited more cracks than HX (Figure 1a) because the HX-a specimen contained additional yttrium (Y) alloying elements that the HX specimen lacked. The notable point is that all the cracks were parallel to the building direction (BD) (Figure 1b). The crack fractions for the HX and HX-a specimens are 1% and 5%, respectively.

Figure 2 shows SEM micrographs of the as-built specimen. At lower magnification, all specimens showed molten pool boundaries and solidification structures, such as grain boundaries and dendrites (Figure 2a,b for HX and Figure 2c,d for HX-a). Figure 2b shows cracks at higher magnification in the HX specimen. In both specimens, the cracks were formed along the grain boundary and appeared in interdendritic regions.

We performed EDS mapping scans at the cracks of the HX and HX-a specimens. EDS analysis revealed SiC carbides and W_6_C formation in the HX specimen (Figure 3a). In the HX-a specimen, EDS mapping at the crack showed the formation of YC (Figure 3b).

Figure 4 shows EDS mapping of the HX-a as-built specimen, which indicates the existence of small Y oxide (yttria) and Si oxide (silica) particles inside the grain. The Electron backscatter diffraction (EBSD) crystallographic orientation maps of both specimens in the as-built condition (Figure 5) revealed columnar grains oriented nearly parallel to the building direction. In the HX specimen, some grains are oriented in the <100> direction and some are in the <101> direction (Figure 5a). On the other hand, the grains in the HX-a specimen are somewhat finer and mostly oriented in the <100> direction (Figure 5b).

#### 3.1.2. ST Specimens

Figure 6 shows SEM micrographs after ST heat treatment. The boundaries of molten pools and dendritic structures disappeared. The HX ST specimen presented equiaxed grain morphology at lower magnification (Figure 6a). That specimen also showed a lot of twins at higher magnifications (Figure 6b). For the HX-a specimen, the ST specimen’s grain morphology was similar to that of the as-built sample (Figure 6c). Two main differences were observed between the HX specimen and the HX-a specimen after ST treatment: in the latter, the grain boundary became thicker with carbide, and those fine carbides formed inside the grain (Figure 6d). In the former, on the other hand, no carbides were observed inside the grain, and the grain boundary was thinner than that of the HX-a ST specimen (Figure 6b). We carried out SEM analysis of the HX-a as-built specimen at the grain boundary; the results are presented in Figure 7a. M_6_C, SiC, and YC were formed at the grain boundary. These carbides at the grain boundary must have pinned the boundary during the solution heat treatment. We carried out FE-SEM analysis at the grain boundary in the HX-a ST specimen. Figure 7b shows the FE-SEM micrograph of the HX-a ST specimen. MC (Si, Y), (Mo, W)_6_C, and Cr_23_C_6_ carbides were formed at the grain boundary. These mainly caused the grain boundary pinning effect to eventually maintain a columnar grain morphology.

Figure 8 shows the IPFs of the HX and HX-a specimens at the ST condition. After the solution heat treatment, the HX specimen showed equiaxed grains and the orientation was random (Figure 8a). Most of the grains have a direction along <101> (Figure 8a). However, the HX-a specimen appeared to be similar to the HX-a as-built specimen (Figure 5b); that is, it had a columnar grain morphology and half of the grains remained along the <100> direction (Figure 8b).

Figure 9a shows EDS mapping of the HX-a ST specimen, which indicates Mo-rich carbides inside the grain. There was also the formation of an oxide of Y and Si-containing C inside the grain (see Figure 9a). In order to find the reason for the accumulation of M_6_C carbides along the interdendritic regions after the solution heat treatment, we performed EDS mapping at the interdendritic areas of the HX-a as-built specimen (Figure 9b); at the interdendritic regions, Mo, Si, C, and O were segregated.

We conducted a creep test along the building direction (vertical specimen) and normal to building directions (horizontal specimen); the creep curves are presented in Figure 10. In the as-built condition, the vertical HX specimen exhibited a creep life of 13.8 h while the HX-a specimen exhibited a creep life 1.46 times higher, 20.2 h (Figure 10a). Moreover, the HX-a revealed a higher creep-rupture elongation (5.7%) than the HX (2.8%). The HX as-built horizontal specimen exhibited longer creep life (3.4 h) than the HX-a horizontal specimen (0.26 h), but the rupture strain was almost the same in both specimens (Figure 10b). Figure 10c shows the creep properties of ST vertical specimens. The HX specimen exhibited a creep life of 3.7 h, while the HX-a specimen showed a creep life eight times higher, 29.6 h. The HX-a showed a higher creep-rupture elongation (15.6%), almost double that of the HX (7.5%). The HX ST horizontal specimen exhibited longer creep life (3.6 h) than the HX-a horizontal specimen (0.26 h), but the creep-rupture elongation was nearly the same in both specimens (Figure 10d).

Figure 11 shows the creep-rupture surfaces. It is evident from Figure 11a,b that the HX and HX-a as-built vertical specimens exhibited elongated grains, which eventually show necking and induce fracturing. By contrast, a rather cleavage-like surface can be observed on the as-built HX and HX-a horizontal specimens (Figure 11c,d, respectively). Obviously, the cracks present perpendicular to the stress axis resulted in the cleavage-like surface along the dendritic structure, indicating brittle behavior and lower ductility.

Figure 12 shows the microstructures of the vertical specimens along the loading direction after the creep test. The HX as-built specimen had large cracks (Figure 12a) along the horizontal grain boundary; these cracks merged and formed significant cracks, resulting in a brittle fracture. We also observed cracks along the grain boundaries in the HX-a as-built specimen (Figure 12c). After solution heat treatment, the HX ST specimen exhibited equiaxed grain morphology, resulting in brittle fracturing of the grain boundary after the creep test (Figure 12b). However, the HX-a ST specimen showed a transgranular fracture (Figure 12d), resulting in a ductile fracture. In addition, in HX-a ST, specimen cracks were aligned parallel to the loading axis, making it difficult for the cracks to propagate normal to the stress axis and resulting in a ductile fracture.

## 4. Discussion

### 4.1. Effect of Segregation on the Hot Cracking Formation

Hastelloy-X is a solid-solution-strengthened Ni-based alloy, and it can exhibit a wide range of melting and solidification temperatures due to its high alloy content. During the SLM process, the base metal adjacent to the fusion zone experiences a range of peak temperatures between the alloy’s liquidus and solidus temperatures. Therefore, this region’s microstructure undergoes partial melting and is described as the partially melted zone (PMZ) region of the HAZ [25]. Liquation cracking occurs at the grain boundaries in the heat-affected zones of welds, also known as HAZ fissuring. During liquation cracking, when the heat melts, low-melting-point phases form on the grain boundaries and at the interdendritic area in the heat-affected zone from a weld run. A liquid film forms on these grain boundaries and interdendritic regions and is pulled apart by the tensile thermal stresses as the weld solidifies. Phases that are likely to be liquated in Ni-based superalloys include MC carbide, M_6_C carbides, Laves phase, and σ-phase [12,26]. The same phenomena were observed in the as-built HX sample (Figure 3a); that is, the segregation of elements such as Si, W, and C brought about the formation of SiC- and W_6_C-type carbides at the grain boundary and interdendritic areas during the solidification process, eventually causing the formation of a crack in the HX specimens (Figure 1a and Figure 3a) [8,17]. The HX-a specimen showed more cracks (Figure 1b) than the HX specimen because its elements were segregated similar to that for additional yttrium (Y) and higher-content Si (Table 1), which resulted in the formation of carbide, mainly MC (M stands for Si, Y) and M_6_C (M stands for W) and caused more cracks to appear (Figure 3b) [26,27]. The solid solubility of Y in the Ni matrix is low and affects crack formation because Y is rejected from the core dendrite to interdendritic regions, causing segregation problems.

### 4.2. Heat Treatment and Anisotropy Effects on Creep Properties

First, we considered the grain morphology effect on creep properties. In the as-built condition, both HX and HX-a specimens exhibited columnar grain formation (Figure 5). Columnar grain formation is a natural phenomenon in additive manufactured materials. This has been proven by many researchers in different alloys [28,29]. The columnar grains are attributed mainly to epitaxial grain growth, a consequence of layer-by-layer formation with rapid heating and cooling during the SLM process [30]. It is well known that materials that have a columnar grain morphology exhibit better creep properties [31]. Although the HX-a as-built specimen had many cracks, it showed better creep properties than the HX specimen. The HX-a vertical specimen’s creep properties showed 1.46 times longer creep life than the HX vertical specimen in the as-built condition (Figure 10a). In addition, yttrium addition in the HX-a specimen formed oxides of Y and Si (Figure 4) and prolonged creep life compared to the HX specimen. The creep properties of the horizontal as-built sample are depicted in Figure 10b. In the horizontal specimens, the HX-a specimen showed an inferior creep life compared to the HX specimen. This is due to the presence of cracks perpendicular to the stress axis (Figure 1b). As a result, the HX specimen having fewer cracks (Figure 1a) showed a longer creep life than the HX-a specimen. However, due to cracks and columnar grain morphology in the as-built condition, anisotropic creep properties prevailed in both HX and HX-a specimens.

The solution treatment changed the microstructures of the HX and HX-a specimens. After the solution heat treatment, the HX specimen showed an equiaxed grain morphology and the orientation became random (Figure 8a). On the other hand, the HX-a specimen maintained a columnar morphology (Figure 8b). The SEM analysis of the HX-a as-built sample at the grain boundary revealed the formation of carbides at the grain boundary, indicating that the grain boundary pinning effect maintained the columnar grain morphology (Figure 7a). FE-SEM analysis was carried out at the grain boundary in the HX-a ST specimen to find phases at the boundary. MC (Si, Y), (Mo, W)_6_C, and Cr_23_C_6_ carbides formed at the grain boundary (Figure 7b). The grain boundary pinning by carbides eventually maintains the columnar grain morphology. Another vital difference between the HX and HX-a ST specimens is the formation of M_6_C carbides inside the HX-a specimen’s grains (Figure 9a). Yttrium promotes a high density of fine Mo-rich carbides and larger oxide inside the grain (Figure 9a). Creep life along the vertical direction of the HX-a ST specimen (29.6 h) was eight times better than that of the HX ST specimen, and creep-rupture elongation became almost double that of the HX ST specimen (Figure 10c). The HX-a ST specimen’s grain morphology was similar to those of the directionally solidified (DS) Ni-based superalloys [29]. Grain boundaries normal to the stress axis are usually the crack initiation sites in conventionally cast superalloys. Therefore, the columnar grain morphology improves the creep life. Thus, the HX-a ST vertical specimen exhibited better creep properties than the HX ST vertical specimen. On the other hand, the HX ST creep test resulted in low creep life and ductility because of the equiaxed grain morphology in the HX ST specimen. There are two additional reasons behind the creep life improvement in the HX-a ST vertical specimen. First, the formation of M_6_C carbides inside the grains in the HX-a specimen (Figure 9a) also affects creep life improvement in the HX-a ST vertical specimen. Second, the Y and Si oxides are stable even at higher temperatures; they also improve creep resistance by hindering dislocation motion; moreover, grain boundary carbides control grain boundary sliding, resulting in the lower creep rate in the HX-a specimen (Figure 13).

Additionally, the formation of continuous carbides at the grain boundary (Figure 14a) results in low ductility; since carbides are brittle phases once the crack nucleates, it propagates rapidly, reducing elongation (Figure 10c). However, from the comparison between Figure 6d and Figure 14b, the discrete carbide increased during the creep test in HX-a ST and improved creep ductility because these carbides (Figure 14b) resist grain boundary sliding and the resultant crack formation. A creep test was also conducted on horizontal solution heat-treated specimens. In the HX-a specimen (Figure 1b), these cracks align perpendicular to the stress axis, and the stress concentration at the crack tip increases. The easy crack propagation results in low creep properties. The HX horizontal specimen showed better creep life than the HX-a horizontal specimen (Figure 10d) since the former had fewer cracks (Figure 1a).

### 4.3. Effects of Oxygen-Induced Grain Boundary Embrittlement on Creep Properties

In powder metallurgy, it is challenging to control the oxygen content of an alloy because alloy powders easily attract oxygen from the atmosphere. In nickel superalloys, oxygen contamination decreases rupture life and ductility in both cast and P/M superalloys. The excess presence of oxygen in an alloy causes a segregation problem at the grain boundaries. This leads to a substantial decrease in the work of separation at the grain boundary, i.e., an increased tendency to form cracks [32]. In addition, the presence of segregated oxygen facilitates vacancy formation at sites close to the grain boundary, which in turn promotes the diffusion of the embrittling atoms to the grain boundary and is expected to result in an increased concentration of the embrittling particles at the grain boundary. The subject of grain-boundary embrittlement by oxygen through environmental contamination has received a great deal of attention and has been reviewed by Woodford and Bricknell [33]. They postulated a link between grain-boundary immobilization and embrittlement. At intermediate temperatures, deformation occurs by grain-boundary sliding and is accommodated by the slip in near-boundary regions and in boundary migration. Oxygen becomes embrittled from boundary immobilization and the absence of grain-boundary accommodation. Several mechanisms have been suggested to be responsible for grain-boundary pinning by oxygen penetration. Two such mechanisms are the segregation of oxygen to grain boundaries and the precipitation of oxygen at sulfides [34]. The presence of oxygen was a prerequisite for SAC (strain age cracking) in René 41, and oxygen segregation to grain boundaries reduces boundary strength. They indicated that oxygen has similar effects on Alloy 718 and Waspaloy [35].

In the present study, the HX as-built specimen contains 115 ppm oxygen. In contrast, the HX-a as-built specimen had an oxygen level of 82 ppm, which is within the 50–100 ppm range at which superalloys experience a significant increment in stress rupture life [36]. We also observed this problem in additively manufactured IN718 and proposed measures to prevent oxygen embrittlement by adding Y to IN718 produced by SLM. The addition of Y improved the creep rupture life and the superalloy’s ductility [37]. In the HX-a specimen formation of Y_2_O_3_ inside the grain (Figure 4), the addition of Y reduces oxygen at the grain boundaries. As a result, the HX-a specimen showed better creep properties despite a lot of vertical cracks. Stabilization of solute oxygen would be one reason why Y addition eventually results in better creep life and rupture elongation in vertical specimens (Figure 10a,c) by preventing grain boundary embrittlement.

## 5. Conclusions

In this study, we investigated the effects of rare earth element Y on the hot cracking and creep properties of the Ni-based superalloy Hastelloy-X processed by selective laser melting. We obtained the following results.

The addition of Y in Hastelloy-X remarkably promoted the formation of cracks. There was segregation of W, Si, C, and Y, causing carbide formation during the SLM process at the cracks. Although fewer cracks formed in the Y-free specimen, W, Si, and C were segregated at the cracks.Although the HX-a sample had many cracks, its creep life was longer than that of the HX sample. This is because the oxygen level was lower (82 ppm) in the HX-a sample and oxygen was stabilized by Y. Most of the oxygen caused the formation of stable Y_2_O_3_ and SiO_2_ oxides, thus eliminating the oxygen embrittlement problem at the grain boundary. In the HX specimen, on the other hand, excessive oxygen (115 ppm) in the alloy causes an oxygen embrittlement problem.After solution treatment, HX-a specimen creep life increased from that in the as-built condition. It was eight times longer than that of the HX ST specimen due to the maintenance of a columnar grain morphology even after solution heat treatment. In addition, due to the formation of M_6_C carbide, SiO_2_ and Y_2_O_3_ oxides improved creep life and ductility compared to the HX ST specimen. The stabilization of solute oxygen is one reason why the addition of Y eventually results in better creep life and rupture elongation of vertical specimens through the prevention of grain boundary embrittlement.In both the HX and HX-a specimens, cracks resulted in anisotropic creep properties. In addition, the presence of a columnar grain morphology in the as-built condition and after solution heat treatment in the HX-a sample also resulted in anisotropic creep properties.

## Figures and Tables

**Figure 1 materials-14-01143-f001:**
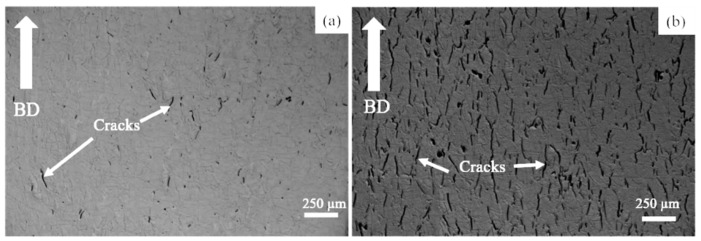
Optical micrographs of as-built (**a**) Hastelloy-X alloys with 0 (HX) and (**b**) and 0.12 (HX-a) yttrium specimens.

**Figure 2 materials-14-01143-f002:**
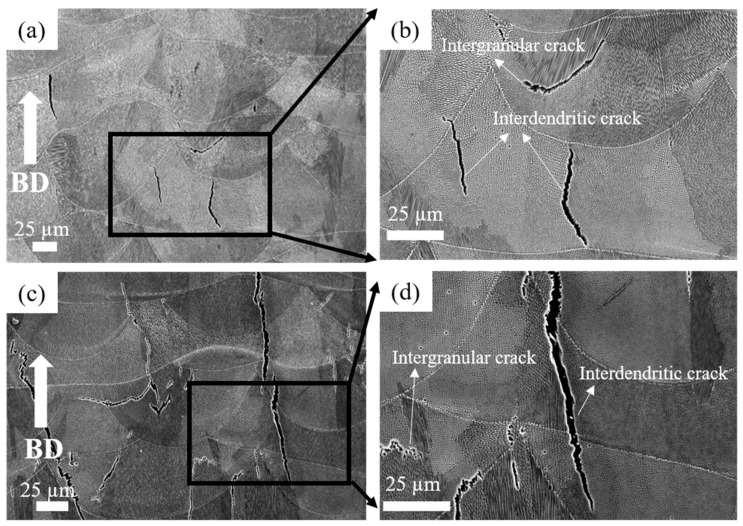
SEM micrographs of as-built HX specimen at (**a**) low magnification and (**b**) high magnification, and HX-a as-built specimen at (**c**) low magnification and (**d**) high magnification.

**Figure 3 materials-14-01143-f003:**
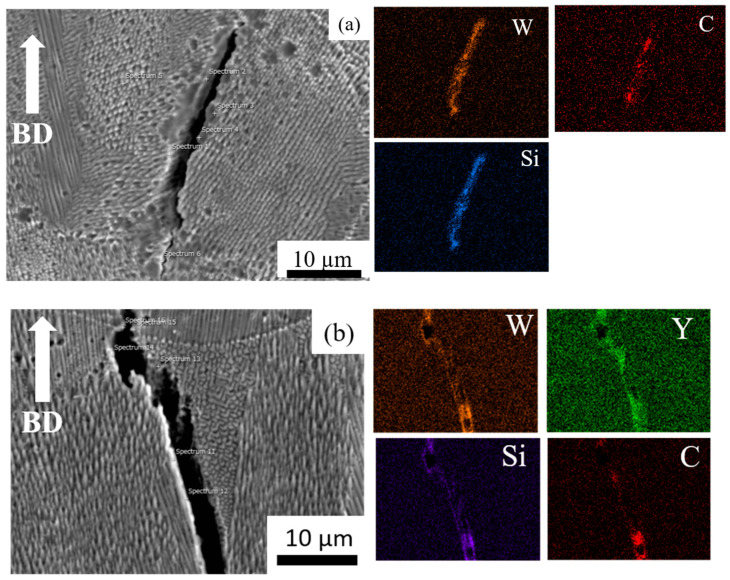
Energy Dispersive X-ray Spectroscopy (EDS or EDX) analysis of (**a**) as-built HX and (**b**) as-built HX-a specimens near the crack.

**Figure 4 materials-14-01143-f004:**
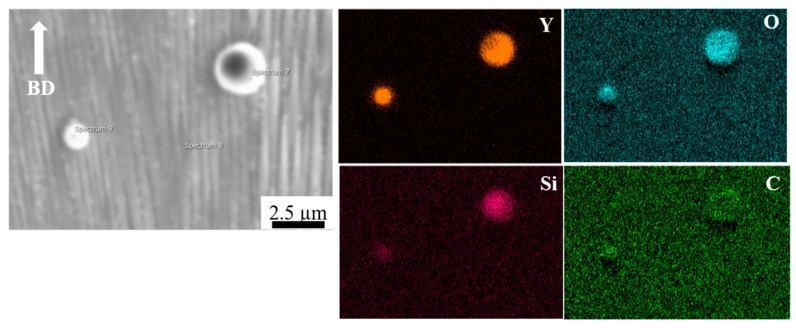
EDS analysis of as-built HX-a specimen in the grain.

**Figure 5 materials-14-01143-f005:**
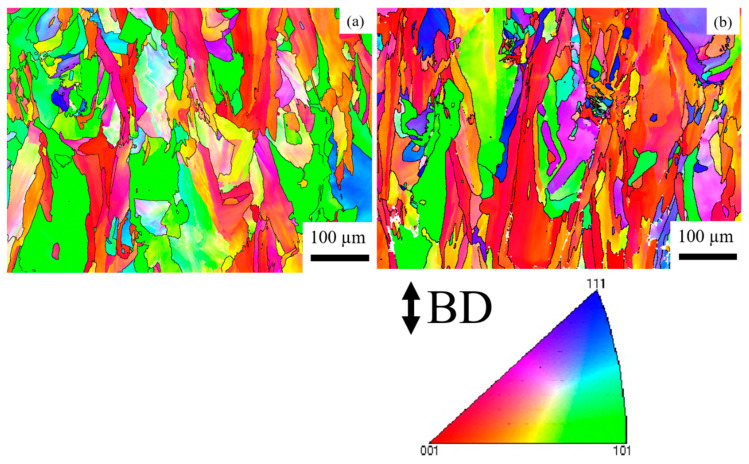
Inverse pole figure (IPF) maps of the as-built (**a**) HX and (**b**) HX-a specimens.

**Figure 6 materials-14-01143-f006:**
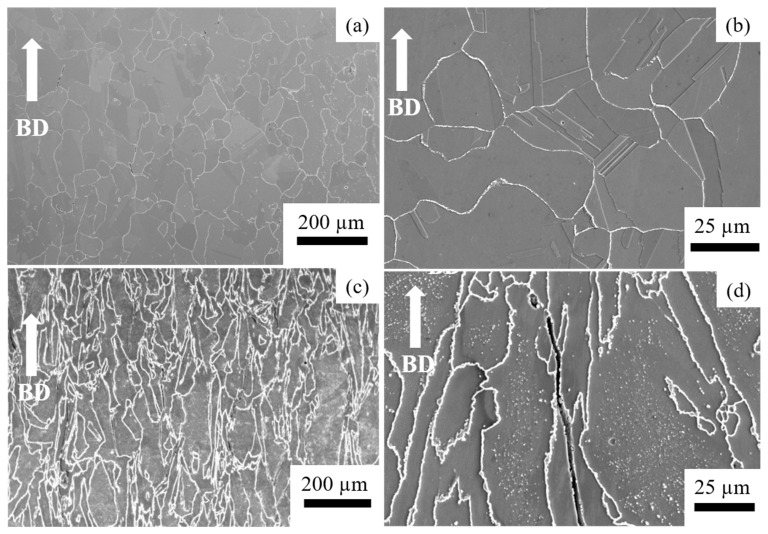
SEM micrographs of the solution heat treatment (ST) HX specimen at (**a**) low magnification and (**b**) high magnification, and HX-a ST specimen at (**c**) low magnification and (**d**) high magnification.

**Figure 7 materials-14-01143-f007:**
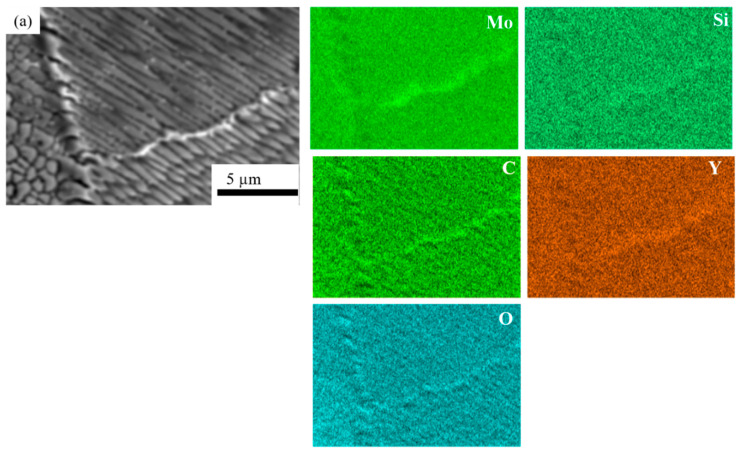
Grain boundary pinning effect in HX-a (**a**) as-built and (**b**) ST specimens at the grain boundary (from the intensity scale bar, the white area shows the highest region of that element).

**Figure 8 materials-14-01143-f008:**
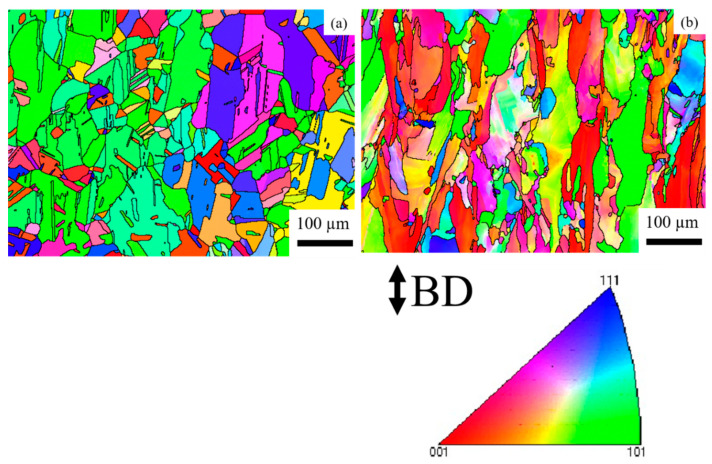
Inverse pole figure (IPF) maps of the ST specimens (**a**) HX and (**b**) HX-a.

**Figure 9 materials-14-01143-f009:**
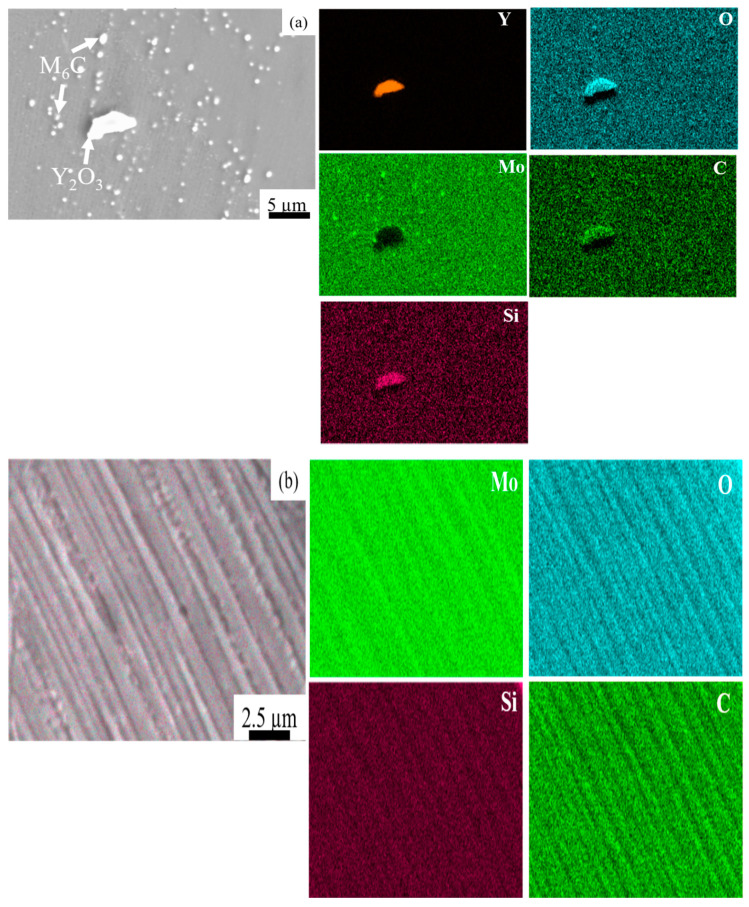
EDS mapping of (**a**) an HX-a ST specimen inside the grain and (**b**) an HX-a as-built sample at the interdendritic regions.

**Figure 10 materials-14-01143-f010:**
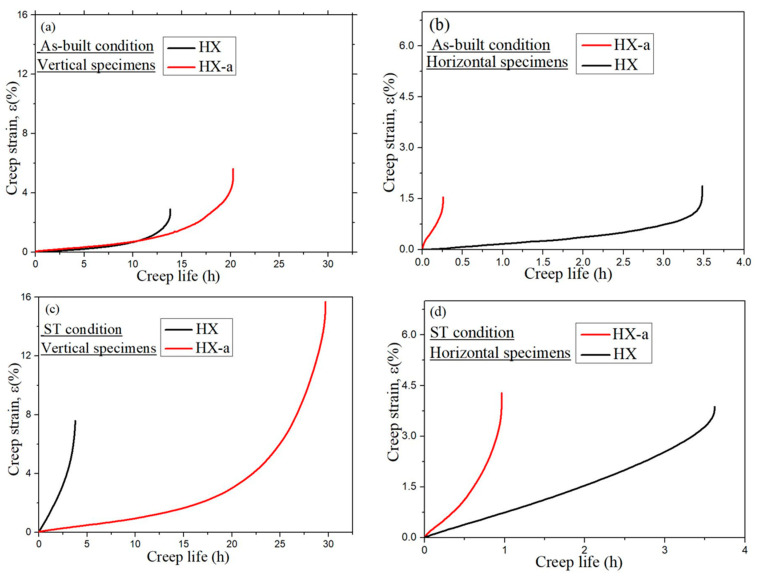
Creep curves of HX and HX-a specimens at 900 °C under 80 MPa: (**a**) as-built vertical, (**b**) as-built horizontal, (**c**) ST vertical, and (**d**) ST horizontal specimens.

**Figure 11 materials-14-01143-f011:**
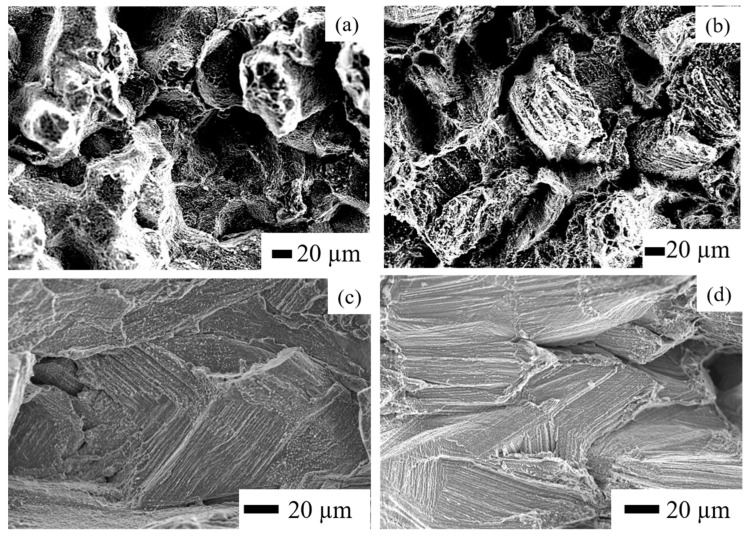
Fracture surfaces after creep tests at 900 °C under 80 MPa, vertical (**a**) HX and (**b**) HX-a as-built specimens, and horizontal (**c**) HX and (**d**) HX-a as-built specimens.

**Figure 12 materials-14-01143-f012:**
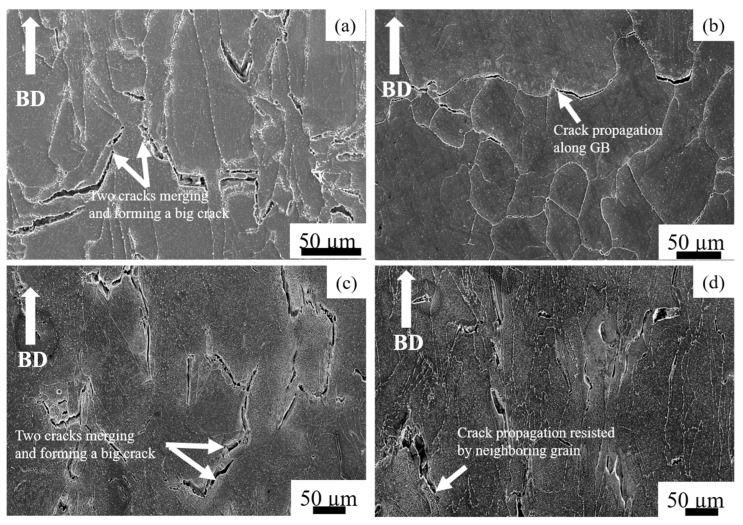
Microstructures in the vertical specimens after creep test (**a**) as-built HX, (**b**) HX ST, (**c**) as-built HX-a, and (**d**) HX-a ST.

**Figure 13 materials-14-01143-f013:**
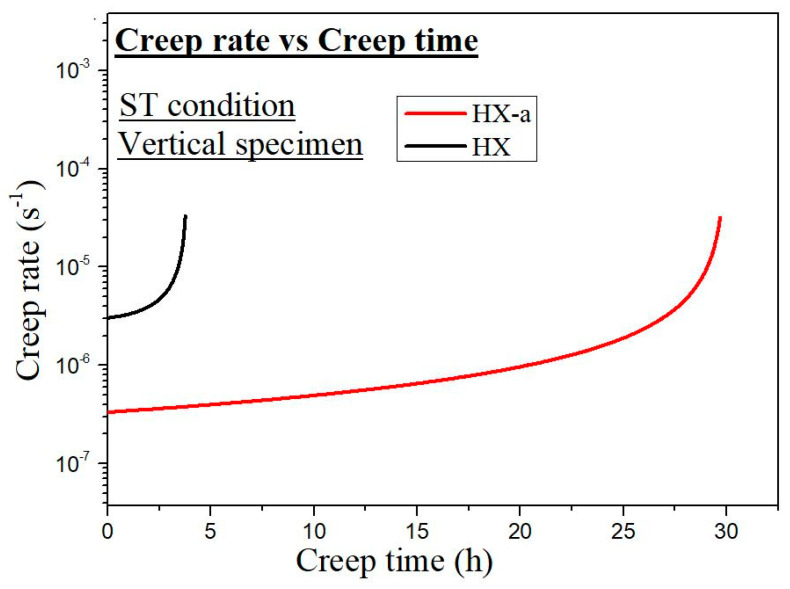
Creep rate vs. creep time curves of ST vertical specimens.

**Figure 14 materials-14-01143-f014:**
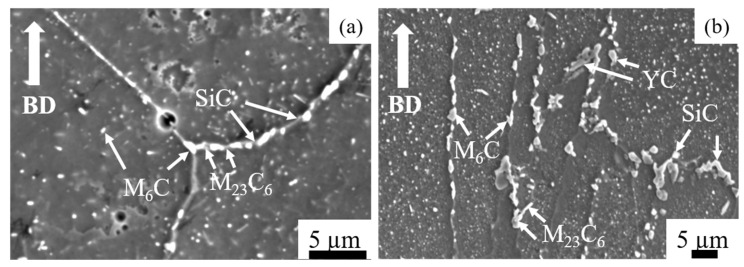
SEM micrographs of the carbides after creep test at 900 °C/80 MPa: (**a**) HX ST and (**b**) HX-a ST.

**Table 1 materials-14-01143-t001:** Chemical composition (mass%).

Specimen	Ni	Cr	Co	Fe	W	Mo	Si	C	O (ppm)	Y
**HX**	Bal.	21.98	1.48	18.04	0.54	8.94	0.09	0.082	115	-
**HX-a**	Bal.	21.86	1.48	18.22	0.53	9.88	0.13	0.101	82	0.12

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
