# Peer review of "Yttrium’s Effect on the Hot Cracking and Creep Properties of a Ni-Based Superalloy Built Up by Additive Manufacturing"

_materials, 2021, doi:10.3390/ma14051143_

Round 1

Reviewer 1 Report

The subject of the manuscript is generally interesting and the text is well organized into paragraphs. The results give important information for future investigations and applications. The introduction provides a good, generalized background of the topic and sufficiently presents the research motivation. However, some explanations, supplementary information, and minor corrections are needed to improve the manuscript’s clarity. 

In the introduction, the Authors gave two alloys that were analyzed - one with the addition, the other without the yttrium addition. Unfortunately, it was not stated that both alloys were analyzed in the as-built condition and after heat treatment. However, the results of the treated samples are given in the following part. Please clarify the information related to the type and condition of tested samples.

Line 23-24: Was it a comparison of samples with and without yttrium addition, both heat-treated? At the beginning and end of a sentence is the expression related to the heat treatment.

Mass % or wt. % - please unify.

There is no explanation in the text of the BD abbreviation used in the figures. 

Line 192: The sentence repeats the information given in the Methods and methods section - unnecessarily in my opinion.

Please add to the text an explanation of what are the samples called “horizontal specimen” and “vertical specimen” (line 191...)

Additionally, slight editorial corrections are needed, for example:

  • in my opinion, the marks (a), (b), (c)... in the figures are disproportionately large and have a different size in each of the figures - please standardize
  • there are no spaces in front of “[“ brackets (reference number) in the text (e.g., lines 38, 39, 40, 52, 328)
  • adapt the text to the journal's requirements, e.g., figure instead of fig., figure caption format, etc. 

Author Response

Thank you for the reviewer’s valuable comments. Modifications are shown with yellow markers.

Reviewer 2 Report

This work discussed the influence of Yttrium on the creep and cracking properties of additively manufactured Nickel alloy. More pre-existing cracks were found in the Yttrium added alloy, but the overall creep performance is better in Yttrium added alloy in the build direction. Detailed discussion was done regarding these phenomena, with microstructure characterization as evidence. In general, the findings are valuable and the discussion is appropriate, and the story is well written. Several minor issues are put forward below.

  1. Only two different yttrium levels (0 and 0.12 mass %) were considered. What would be the influence of the level of yttrium on the properties? This question is of practical interest. The authors are invited to give a qualitative discussion, although additional experiment is not necessary. For instance, the authors attributed the better creep performance to reduced oxygen content and so less embrittlement, then would it be better to further increase yttrium content/decrease oxygen?
  2. The definition of vertical and horizontal specimens is a bit misleading: vertical – parallel to the build direction; horizontal – vertical to the build direction. It took some effort to figure out.
  3. In Fig.11, it is very interesting that the final creep strain for the horizontal specimens are so close. The authors are invited to give a discussion.
  4. Table 2 is redundant and can be explained by a single sentence in the text.
  5. What is the practical value of the yttrium added alloy? It is better in the building direction but worse in the perpendicular direction.

Author Response

(The authors gave the same response as above.)

Reviewer 3 Report

The manuscript “Yttrium’s effect on the hot cracking and creep properties of a Ni-based superalloy built up by additive manufacturing” presents experimental results that are of interest to the scientific community. However, I cannot recommend the manuscript for publication in the present form. In general, in the first part (from abstract to the microstructure results) it is not well written, some sentences are not clear. The second part of the manuscript (including creep tests, discussion and conclusions) is much better. Overall, a major revision including English correction by a native speaker is needed.

Line 13 – parentheses around wt.% are superfluous, could be omitted.

Line 14 – 16: Sentence: “More cracks formed in the yttrium-added specimen mainly because of segregation such as that between yttrium (Y), W, Si, and C, resulting in the formation of carbides, mainly (Si, Y)C, and W6C, thereby causing more cracks to form at the grain boundary and interdendritic areas.” is too long and not clear. It should be reformulated. E.g.: More cracks formed in the yttrium-added specimen because of segregation of Y, W, Si, and C, resulting in formation of carbides, mainly (Si,Y)C and W6C. The cracks were located at the grain boundaries and in interdendritic areas.

Line 16-18. Sentence: “In comparison, as-built Hastelloy-X exhibited fewer cracks, mainly due to the segregation of Si, W, and C resulting in SiC and W6C-type carbides at the grain boundary and interdendritic regions.” says practically the same as the previous one, but for the opposite, i.e. fewer cracks due to SiC and W6C carbides. Is the only difference (Si,Y)C carbide for more cracks and SiC carbide for fewer cracks? This is not clear and it should be clarified and reformulated.

Line 22 – misprint, s as a plural of grain missing: … inside the grains.

Line 23 – misprint – negative verb “NOT” missing, or wrong formulation: “After solution treatment, the Y-added specimen's creep life was eight times longer than that of the solution-treated specimen.” Should be … than that of the not solution-treated specimen, or reformulated “… than that of the as-built specimen”.

Line 24: sentence: “This was mainly because of the columnar grain morphology resulting in the formation of M6C carbides, Y2O3, and SiO2.” is not correct. The HX-a sample before and after solution heat treatment have the same columnar grain morphology (cf. Fig. 6 and Fig. 9, which are practically the same). In solution treated (ST) samples there are carbides on grain boundaries and oxides inside the grains, but the columnar grain morphology is already present in the Hx-a as-built material and does not result from carbide and oxide formation during ST. It should be reformulated accordingly.

Line 36: abbreviation of the EOS company is not defined. The specification of the SLM device is not relevant in the introduction, so EOS could be omitted. Furthermore, a reference is missing at the end of the sentence: …. technology [xx].

Line 38 – 40 spaces missing before references: … process[8],  should be process [8] and so on.

Line 59: Formulation problem: Zhou et al.'s work on nickel-based superalloys showed that the optimum level of yttrium in nickel-based alloys improves the stress-rupture property [20] and the oxidation resistance of nickel-based superalloy [21], [22].” In the sentence there is twice “nickel-based superalloys”. Once it should be omitted.

Line 72: please, replace “with” by “and”.

Line 76. Table 1. There are mass.% in the table heading. In the abstract and on line 69 there is wt.%. It should be unified throughout the paper.

Line 80: The sentence: “For the microstructural observation, specimens were cut using the electro-discharge wire cutting machine from the 3.1 mm thick slabs.” is superfluous. It could be omitted, it is already said before. only “Specimens for the microstructural observation” could be joined to the following sentence: “Specimens for the microstructural observation were polished …”

Line 85 to 92 – the description of the equipment is not clear. It should be reformulated. There are two SEM microscopes used. FE gun SEM is not attached to EDS. FE SEM is equipped with EDS. If you give the manufacturers of the devices, you should give it also for EPMA, which is a stand-alone device, or there is a WDS system attached to a SEM.

Line 101: “We identified significantly lower residual porosity in both specimens.” There is a problem of definition of cracks and residual porosity. In Fig. 1a the cracks are very small and difficult to be distinguished from the residual porosity. Maybe you could use aspect ratio (porosity – mostly round pores with aspect ratio 1 to 2.5, cracks – aspect ratio > 3)? Furthermore, the sentence gives a comparison: … should be “.. significantly lower residual porosity than …” The difference between cracks and residual porosity should be defined and the sentence should be reformulated accordingly to be clear.

Line 103: “Fig. 1. Optical micrographs graphs of as-built (a) HX and (b) HX-a specimens.” The word “graphs” is superfluous and should be omitted.

Line 105: Misprint: Full stop missing: should be “Table 2. Crack area fractions”

Line 107: The sentence: “At lower magnification, all specimens showed molten pool boundaries and crack formation due to laser beam tracing during the SLM process with solidification structures and crack distribution ..” is not clear. Perhaps it could be:” At lower magnification, all specimens showed molten pool boundaries and solidification structures, such as grain boundaries and dendrites. In both specimens, the cracks formed during the SLM process along grain boundaries and between dendrites.” And the following text – line 110 to 113: “Figure 2b shows cracks at higher magnification in the HX specimen. The cracks were formed along the grain boundary, and interdendritic regions appeared. The same observation was made in the HX-a specimen at the higher magnification; all the cracks were also found along the grain boundaries and interdendritic regions (Fig. 2d).” could be omitted.

Line 110: The sentence: “The cracks were formed along the grain boundary, and interdendritic regions appeared.” Is not clear. Perhaps it could be: “The cracks were formed along the grain boundaries, and in interdendritic regions.”

Line 128: Fig. 4 – Why there is only an EPMA analysis of Si in the Y-containing (HX-a) specimen and not also for the specimen without Y (HX)? From Fig. 3a it is clear that Si is also present close to the crack in the specimen without Y. This EPMA result does not give any new information and Fig. 4 should be omitted. Could the authors comment on this?

Line 137: Formulation problem: “The vertical direction is parallel to the building direction. In the HX specimen, some grains are oriented in the <100> direction (Fig. 6a). On the other hand, the HX-a specimen also had <100>-oriented grains, which included columnar and fine grains (Fig. 6b).” If you use “on the other hand”, you should present a different meaning and not the same. Perhaps the whole part describing Fig. 6 could be: “The EBSD crystallographic orientation maps of both specimens in the as-built condition (Fig. 6) revealed columnar grains oriented nearly parallel to the building direction. In the HX specimen, some grains are oriented in the <100> direction, some in <101> direction (Fig. 6a). On the other hand, the grains in the HX-a specimen are somewhat finer, and mostly oriented in <100> direction (Fig. 6b)”.

Line 150: Formulation problem: “Figure 7 shows SEM micrographs after ST heat treatment. Neither specimen showed molten pools or dendritic structures.” Perhaps it would be better: “Figure 7 shows SEM micrographs after ST heat treatment. The boundaries of molten pools and dendritic structures disappeared”.

Line 153 – “For the HX-a specimen, the ST specimen's microstructure was similar to that of the as-built sample (Fig. 7c).” I do not see any similarity with Fig. 2 with molten pools, and dendrites. Perhaps you could compare the grain structure before and after ST, i.e. Fig. 7c,d with Fig. 6b, but the microstructure is more complex. It should be reformulated.

Line 166 – The contrast of Mo and Si on the EDS maps (Fig. 8a) is very faint. I suggest to perform a FE SEM analysis for the as-built specimen HX-a as for the specimen HX-a ST to have better results, which would be comparable with Fig. 8b.

Line 174 – A contradiction: The statement: “After the solution heat treatment, the HX specimen showed equiaxed grains and the orientation was random (Fig. 9a). Most of the grains have a direction along <101> (Fig. 9a).” is not correct – it cannot be random and at the same time most of the grains oriented along <101>. Furthermore, the grains are not equiaxed, but somewhat elongated in the building direction. It should be reformulated.

Line 178 – Mo-rich carbides inside the grain. According to the EDS maps in the Fig. 10a the particle is SiC, and not Mo-rich. If the map shows the brighter colour as rich in the element, black one in the Mo map means no element.

Line 227 – Fig. 13b – text in the figure not correct. There is “Crack propagation through GB” and should be “Crack propagation along GB”.

Line 238 – wrong preposition: cracking occurs at grain boundaries (not in GB)

Author Response

(The authors gave the same response as above.)

Reviewer 4 Report

The paper is focused on hot cracking and creep live of Ni-based superalloy processed by additive manufacturing. Main output of the paper is that the addition of Yttrium could result in an improvement of the creep life and rupture elongation of Hastelloy-X processed by selective laser melting.

Paper contains interesting (and many) results. However, the reasoning is frequently too straightforward so the causes and consequences are often confused.

Two example for all:

Line 98 “The HX-a specimen exhibited more cracks than HX (Fig. 1a) because the HX-a specimen contained additional yttrium (Y) alloying elements that the HX specimen lacked.” Conjunction “because” imply why it is so, which is not explained in this statement…

Line 222 “However, the HX-a ST specimen showed a transgranular cleavage fracture (Fig. 13d) resulting in a ductile fracture.” - again, the statement that transgranular cleavage results in a ductile fracture is too straightforward and very confusing (I understand why it is so, but it should be better explained in the paper...).

In my opinion, the structure of the paper should be improved. Sometimes, it is hard to follow as the negative and positive effects of Y-addition are mixed together (Y remarkably promoted the formation of cracks, … creep life was longer, … this is because the oxygen level was lower, … due to the maintenance of columnar grain morphology etc).

Authors should also focus on the reasons (or should better explain) why:

i) the Y promoted the formation of cracks (and if it could be eliminated maintaining positive effect of Y-addition)

ii) the columnar grain morphology was maintained after solution treatment in Y-added samples.

To conclude, the paper should be rewritten in a more comprehensive way before it can be recommended for publication.

Author Response

(The authors gave the same response as above.)

Round 2

Reviewer 3 Report

The authors took into account all the reviewer´s suggestions. The paper is now much better. I recommend it for publication with very minor stylistic correction on line 105: There is: "In both specimens, the cracks were formed along the grain boundary, and interdendritic regions appeared." It should be better: ... grain boundaries, and appeared in interdentritic regions."

Author Response

(The authors gave the same response as above.)

Reviewer 4 Report

Authors did some effort to improve the quality of the paper, althought the structure of the paper was practically not changed (according to previous comments), so it is still quite hard to follow the paper.

English needs revision (some phrases are not grammatically correct). Seek the help of a native English speaker.

Author Response

(The authors gave the same response as above.)
